# Multi-mapping Image-to-Image Translation via Learning Disentanglement

**Xiaoming Yu**[1,2]**, Yuanqi Chen**[1,2]**, Thomas Li**[1,3]**, Shan Liu**[4]**, and Ge Li** ✉[1,2]

[1]School of Electronics and Computer Engineering, Peking University  [2]Peng Cheng Laboratory
[3]Advanced Institute of Information Technology, Peking University  [4]Tencent America
xiaomingyu@pku.edu.cn, cyq373@pku.edu.cn
tli@aiit.org.cn, shanl@tencent.com, geli@ece.pku.edu.cn

## Abstract

Recent advances of image-to-image translation focus on learning the one-to-many mapping from two aspects: multi-modal translation and multi-domain translation. However, the existing methods only consider one of the two perspectives, which makes them unable to solve each other's problem. To address this issue, we propose a novel unified model, which bridges these two objectives. First, we disentangle the input images into the latent representations by an encoder-decoder architecture with a conditional adversarial training in the feature space. Then, we encourage the generator to learn multi-mappings by a random cross-domain translation. As a result, we can manipulate different parts of the latent representations to perform multi-modal and multi-domain translations simultaneously. Experiments demonstrate that our method outperforms state-of-the-art methods. Code will be available at https://github.com/Xiaoming-Yu/DMIT.

## 1  Introduction

Image-to-image (I2I) translation is a broad concept that aims to translate images from one domain to another. Many computer vision and image processing problems can be handled in this framework, *e.g.* image colorization [16], image inpainting [39], style transfer [45], *etc*. Previous works [16, 45, 40, 18, 24] present the impressive results on the task with deterministic one-to-one mapping, but suffer from mode collapse when the outputs correspond to multiple possibilities. For example, in the season transfer task, as shown in Fig. 1, a summer image may correspond to multiple winter scenes with different styles of lighting, sky, and snow. To tackle this problem and generalize the applicable scenarios of I2I, recent studies focus on one-to-many translation and explore the problem from two perspectives: multi-domain translation [20, 3, 25], and multi-modal translation [46, 22, 15, 42, 39].

The multi-domain translation aims to learn mappings between each domain and other domains. Under a single unified framework, recent works realize the translation among multiple domains. However, between the two domains, what these methods have learned are still deterministic one-to-one mappings, thus they fail to capture the multi-modal nature of the image distribution within the image domain. Another line of works is the multi-modal translation. BicycleGAN [46] achieves the one-to-many mapping between the source domain and the target domain by combining the objective of cVAE-GAN [21] and cLR-GAN [2, 5, 7]. MUNIT [15] and DRIT [22] extend the method to learn two one-to-many mappings between the two image domains in an unsupervised setting, i.e., domain A to domain B and vice versa. While capable of generating diverse and realistic translation outputs, these methods are limited when there are multiple image domains to be translated. In order to adapt to the new task, the domain-specific encoder-decoder architecture in these methods needs to be duplicated to the number of image domains. Moreover, they assume that there is no correlation of the styles between domains, while we argue that they could be aligned as shown in Fig. 1. Besides,

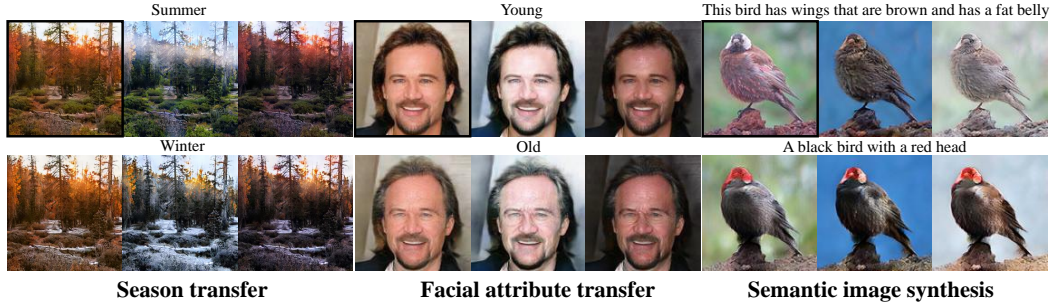

| Summer | Young | This bird has wings that are brown and has a fat belly |
|---|---|---|
| Winter | Old | A black bird with a red head |
| **Season transfer** | **Facial attribute transfer** | **Semantic image synthesis** |

Figure 1: **Multi-mapping image-to-image translation.** The images with a black border are the input images, and other images are generated by our method. The images on the same column have the same style, which indicates that the styles between image domains could be aligned.

existing one-to-many mapping methods usually assume the state of the domain is finite and discrete, which limits their application scenarios.

In this paper, we focus on bridging the objectives of multi-domain translation and multi-modal translation with an unsupervised unified framework. For clarity, we refer our task to as multi-mapping translation. Simultaneous modeling for these two problems not only makes the framework more efficient but also encourages the model to learn efficient representations for diverse translations.

To instantiate the idea, as shown in Fig. 2(d), we assume that the images can be disentangled into two latent representation spaces: a content space $\mathcal{C}$ and a style space $\mathcal{S}$, and propose an encoder-decoder architecture to learn the disentangled representations. Our assumption is developed by the shared latent space assumption [24], but we disentangle the latent space into two separate parts to model the multi-modal distribution and to achieve cross-domain translation. Unlike partially-shared latent space assumption [15, 22], that treats style information as domain-specific, the styles between image domains are aligned in our assumption, as shown in Fig. 1. Specifically, the style representations in this work are low-dimensional vectors which do not contain spatial information and hence can only control the global appearance of the outputs. By using a unified style encoder to learn style representations and thus fully utilizing samples of all image domains, the sample space of our style representation is denser than that learned from only one specific image domain. As for content representations, they are feature maps capturing the spatial structure information across domains. To mitigate the effects of distribution shift among domains, we eliminate domain-specific information in content representations via conditional adversarial learning. To achieve multi-mapping translation using a single unified decoder, we concatenate the disentangled style representations with the target domain label, then adopt the style-based injection method to render the content representations to our desired outputs. Through learning the inverse mapping of disentanglement, we can change the domain label to translate an image to the specific domain or modify the style representation to produce multi-modal outputs. Furthermore, we can extend our framework to a more challenging task of semantic image synthesis whose domains can be considered as an uncountable set and cannot be modeled by existing I2I approaches.

The contributions of this work are summarized as follows:

- We introduce an unsupervised unified multi-mapping framework, which unites the objectives of multi-domain and multi-modal translations.
- By aligning latent representations among image domains, our model is efficient in learning disentanglement and performing finer image translation.
- Experimental results show our model is superior to the state-of-the-art methods.

## 2   Related Work

**Image-to-image translation.** The problem of I2I is first defined by Isola *et al*. [16]. Based on the generative adversarial networks [11, 27], they propose a general-purpose framework (pix2pix) to handle I2I. To get rid of the constraint of paired data in pix2pix, [45, 40, 18] utilize the cycle-

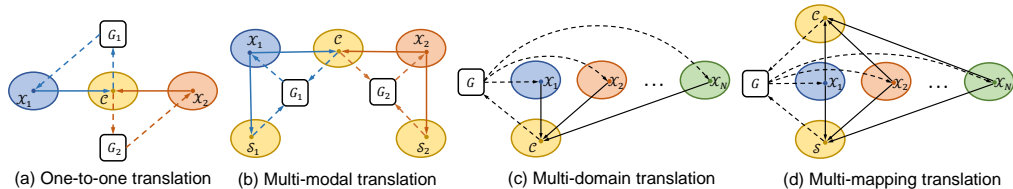

(a) One-to-one translation    (b) Multi-modal translation    (c) Multi-domain translation    (d) Multi-mapping translation

Figure 2: **Comparisons of unsupervised I2I translation methods.** Denote $\mathcal{X}_k$ as the k-th image domain. The solid lines and dashed lines represent the flow of encoder and generator respectively. The lines with the same color indicate they belong to the same module.

Table 1: Comparisons with recent works on unsupervised image-to-image translation

| | Multi-modal translation | Multi-domain translation | Multi-mapping translation | Unified structure | Feature disentanglement | Representation alignment |
|---|---|---|---|---|---|---|
| UNIT | - | - | - | - | - | ✓ |
| StarGAN | - | ✓ | - | ✓ | - | - |
| MUNIT | ✓ | - | - | - | ✓ | Partial |
| DRIT | ✓ | - | - | - | ✓ | Partial |
| SingleGAN | ✓ | ✓ | - | - | - | - |
| Ours | ✓ | ✓ | ✓ | ✓ | ✓ | ✓ |

consistency for the stability of training. UNIT [24] assumes a shared latent space for two image domains. It achieves unsupervised translation by learning the bijection between latent and image spaces using two generators. However, these methods only learn the one-to-one mapping between two domains and thus produce deterministic output for an input image. Recent studies focus on multi-domain translation [20, 3, 42, 25] and multi-modal translation [46, 39, 22, 15, 39, 42, 31]. Unfortunately, neither multi-modal translation nor multi-domain translation considers the other's scenario, which makes them unable to solve the problem of each other. Table 1 shows a feature-by-feature comparison among various unsupervised I2I models. Different from the aforementioned methods, we explore a combination of these two problems rather than separation, which makes our model more efficient and general purpose. Concurrent with our work, several independent researches [4, 33, 37] also tackle the multi-mapping problem from different perspectives.

**Representation disentanglement.** To achieve a finer manipulation in image generation, disentangling the factors of data variation has attracted a great amount of attention [19, 13, 2]. Some previous works [20, 25] aim to learn domain-invariant representations from data across multiple domains, then generate different realistic versions of an input image by varying the domain labels. Others [22, 15, 10] focus on disentangling the images into domain-invariant and domain-specific representations to facilitate learning diverse cross-domain mappings. Inspired by these works, we attempt to disentangle the images into solely independent parts: content and style. Moreover, we align these representations among image domains, which allows us to utilize rich content and style from different domains and manipulate the translation in finer detail.

**Semantic image synthesis.** The goal of semantic image synthesis is to generate an image to match the given text while retaining the irrelevant information from the input image. Dong *et al.* [6] train a conditional GAN to synthesize a manipulated version of the image given an original image and a target text description. To preserve text-irrelevant contents of the original image, Paired-D GAN [26] proposes to model the foreground and background distribution with different discriminators. TAGAN [30] introduces a text-adaptive discriminator to pay attention to the regions that correspond to the given text. In this work, we treat the image set with the same text description as an image domain. Thus the domains are countless and each domain contains very few images in the training set. Benefit from the unified framework and the representation alignment among different domains, we can tackle this problem in our unified multi-mapping framework.

## 3   Proposed Method

Let $\mathcal{X} = \bigcup_{k=1}^{N} \mathcal{X}_k \subset \mathbb{R}^{H \times W \times 3}$ be an image set that contains all possible images of $N$ different domains. We assume that the images can be disentangled to two latent representations $(\mathcal{C}, \mathcal{S})$. $\mathcal{C}$ is

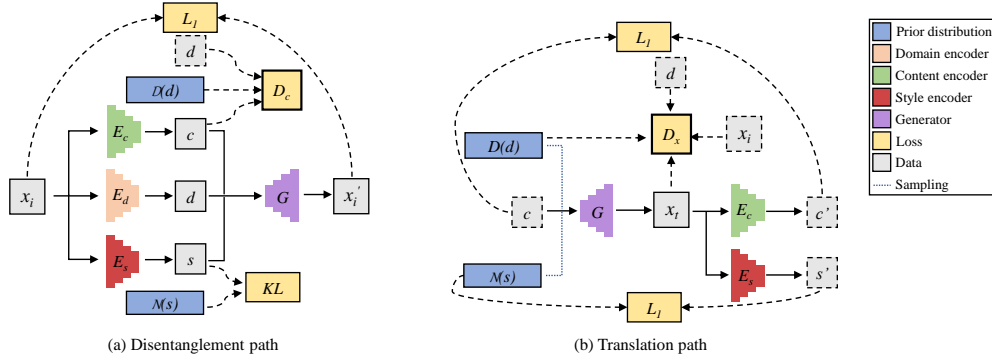

(a) Disentanglement path          (b) Translation path

Figure 3: **Overview.** (a) The disentanglement path learns the bijective mapping between the disentangled representations and the input image. (b) The translation path encourages to generate diverse outputs with possible styles in different domains.

the set of contents excluded from the variation among domains and styles, and $\mathcal{S}$ is the set of styles that is the rendering of the contents. Our goal is to train a unified model that learns multi-mappings among multiple domains and styles. To achieve this goal, we also define $\mathcal{D}$ as a set of domain labels and treat $\mathcal{D}$ as another disentangled representations of the images. Then we propose to learn mapping functions between images and disentangled representations $\mathcal{X} \rightleftharpoons (\mathcal{C}, \mathcal{S}, \mathcal{D})$.

As illustrated in Fig. 3(a), we introduce the content encoder $E_c : \mathcal{X} \rightarrow \mathcal{C}$ that maps an input image to its content, and the encoder style $E_s : \mathcal{X} \rightarrow \mathcal{S}$ that extracts the style of the input image. To unify the formulation, We also denote the determined mapping function between $\mathcal{X}$ and $\mathcal{D}$ as the domain label encoder $E_d : \mathcal{X} \rightarrow \mathcal{D}$ which is organized as a dictionary[1] and extracts the domain label from the input image. The inversely disentangled mapping is formulated as the generator $G : (\mathcal{C}, \mathcal{S}, \mathcal{D}) \rightarrow \mathcal{X}$. As a result, with any desired style $s \in \mathcal{S}$ and domain label $d \in \mathcal{D}$, we can translate an input image $x_i \in \mathcal{X}$ to the corresponding target $x_t \in \mathcal{X}$

$$x_t = G(E_c(x_i), s, d). \tag{1}$$

## 3.1   Network Architecture

**Encoder.** The content encoder $E_c$ is a fully convolutional network that encode the input image to the spatial feature map $c$. Since the small output stride used in $E_c$, $c$ retains rich spatial structure information of input image. The style encoder $E_s$ consists of several residual blocks followed by global average pooling and fully connected layers. By global average pooling, $E_s$ removes the structure information of input and extract the statistical characteristics to represent the input style [9]. The final style representation $s$ are constructed as a low-dimensional vector by the reparameterization trick [19].

**Generator.** Motivated by recent style-based methods [8, 14, 17, 15, 42], we adopt a style-based generator $G$ to simultaneous model for multi-domain and multi-modal translations. Specifically, the generator $G$ consists of several residual blocks followed by several deconvolutional layers. Each convolution layer in residual blocks is equipped with CBIN [42, 43] for information injection.

**Discriminator.** Unlike previous works [22, 15, 42] that apply different discriminators for different image domains, we propose to adopt a unified conditional discriminator for different domains. Since the large distribution shift between image domains in I2I, it is challenging to use a unified discriminator. Inspired by the style-based generator, we apply CBIN to the discriminator to extend the capacity of our model. For more details of our network, we refer the reader to our supplementary materials.

## 3.2   Learning Strategy

Our proposed method encourages the bijective mapping between the image and the latent representations while learning disentanglement. Fig. 3 presents an overview of our model, whose learning

process can be separated into disentanglement path and translation path. The disentanglement path can be considered as an encoder-decoder architecture that uses conditional adversarial training on the latent space. Here we enforce the encoders to encode the image into the disentangled representations, which can be mapped back to the input image by the conditional generator. The translation path enforces the generator to capture the full distribution of possible outputs by a random cross-domain translation.

**Disentanglement path.** To disentangle the latent representations from image $x_i$, we adopt cVAE [34] as the base structure. To align the style representations across visual domains and constrain the information of the styles [1], we encourage the distribution of styles of all domains to be as close as possible to a prior distribution.

$$\mathcal{L}_{cVAE} = \lambda_{KL}\mathbb{E}_{x_i \sim \mathcal{X}}[KL(E_s(x_i)||q(s)] + \lambda_{rec}\mathbb{E}_{x_i \sim \mathcal{X}}[\|G(E_c(x_i), E_s(x_i), E_d(x_i)) - x_i\|_1]. \quad (2)$$

To enable stochastic sampling at test time, we choose the prior distribution $q(s)$ to be a standard Gaussian distribution $\mathcal{N}(0, I)$. As for the content representations, we propose to perform conditional adversarial training in the content space to address the distribution shift issue of the contents among domains. This process encourages $E_c$ to exclude the information of the domain $d$ in content $c$

$$\mathcal{L}_{GAN}^c = \mathbb{E}_{x_i \sim \mathcal{X}}[\log(D_c(E_c(x_i), E_d(x_i))) + \mathbb{E}_{d \sim (\mathcal{D} - \{E_d(x_i)\})}[\log(1 - D_c(E_c(x_i), d))]]. \quad (3)$$

the overall loss of the disentanglement path is

$$\mathcal{L}_{D-Path} = \mathcal{L}_{cVAE} + \mathcal{L}_{GAN}^c. \quad (4)$$

**Translation path.** The disentanglement path encourages the model to learn the content $c$ and the style $s$ with a prior distribution. But it leaves two issues to be solved: First, limited by the number of training data and the optimization of KL loss, the generator $G$ may sample only a subset of $\mathcal{S}$ and generate the images with specific domain labels in the training stage [35]. It may lead to poor generations when sampling $s$ in the prior distribution $\mathcal{N}$ and $d$ that does not match the test image, as discussed in [46]. Second, the above training process lacks efficient incentives for the use of styles, which would result in low diversity of the generated images. To overcome these issues and encourage our generator to capture a complete distribution of outputs, we first propose to randomly sample domain labels and styles in the prior distributions, in order to cover the whole sampling space at training time. Then we introduce the latent regression [2, 46] to force the generator to utilize the style vector. The regression can also be applied to the content $c$ to separate the style $s$ from $c$. Thus the latent regression can be written as

$$\mathcal{L}_{reg} = \mathbb{E}_{\substack{c \sim \mathcal{C} \\ s \sim \mathcal{N} \\ d \sim \mathcal{D}}}[\|E_s(G(c, s, d)) - s\|_1] + \mathbb{E}_{\substack{c \sim \mathcal{C} \\ s \sim \mathcal{N} \\ d \sim \mathcal{D}}}[\|E_c(G(c, s, d)) - c\|_1]. \quad (5)$$

To match the distribution of generated images to the real data with sampling domain labels and styles, we employ conditional adversarial training in the pixel space

$$\mathcal{L}_{GAN}^x = \mathbb{E}_{x_i \sim \mathcal{X}}[\log(D_x(x_i, E_d(x_i))) + \mathbb{E}_{d \sim (\mathcal{D} - \{E_d(x_i)\})}[\frac{1}{2}\log(1 - D_x(x_i, d))$$
$$+ \mathbb{E}_{s \sim \mathcal{N}}[\frac{1}{2}\log(1 - D_x(G(E_c(x_i), s, d), d))]]]. \quad (6)$$

Note that we also discriminate the pair of real image $x_i$ and mismatched target domain label $d$, in order to encourage the generator to generate images that correspond to the given domain label. The final objective of the translation is

$$\mathcal{L}_{T-Path} = \lambda_{reg}\mathcal{L}_{reg} + \mathcal{L}_{GAN}^x. \quad (7)$$

By combining both training paths, the full objective function of our model is

$$\min_{G, E_c, E_s} \max_{D_c, D_x} \mathcal{L}_{D-Path} + \mathcal{L}_{T-Path}. \quad (8)$$

## 4 Experiments

We compare our approach against recent one-to-many mapping models in two tasks, including season transfer and semantic image synthesis. For brevity, we refer to our method, Disentanglement for Multi-mapping Image-to-Image Translation, as `DMIT`. In the supplementary material, we provide additional visual results and extend our model to facial attribute transfer [23] and sketch-to-photo [41].

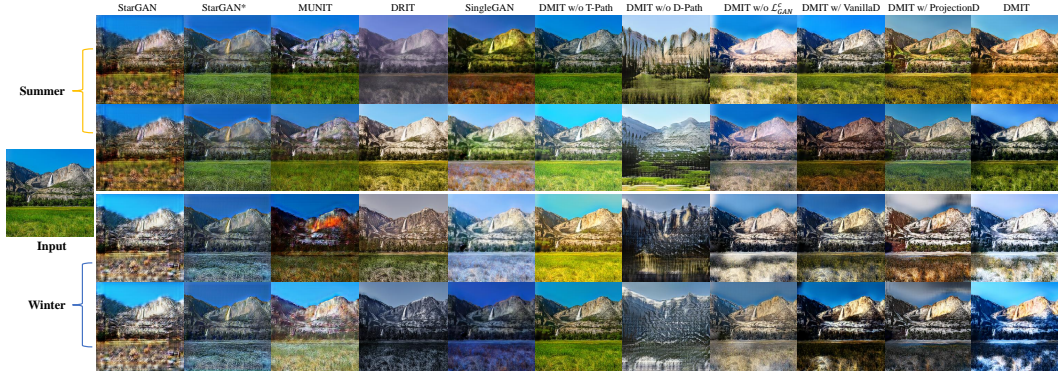

Figure 4: Qualitative comparison of season transfer. The first column shows the input image. Each of the remaining columns shows four outputs with the specified season from a method. Each image pair for the specified season reflects the diversity within the domain.

## 4.1 Datasets

**Yosemite summer ↔ winter.** The unpaired dataset is provided by Zhu *et al*. [45] for evaluating unsupervised I2I methods. We use the default image size 256×256 and training set in all experiments. The domain label(summer/winter) is organized as a one-hot vector.
**CUB.** The Caltech-UCSD Birds (CUB) [36] dataset contains 200 bird species with 11,788 images that each have 10 text captions [32]. We preprocess the CUB dataset according to the method in [38]. The captions are encoded as the domain labels by the pretrained text encoder proposed in [38].

## 4.2 Season Transfer

Season transfer is a coarse-grained translation task that aims to learn the mapping between summer and winter. We compare our method against five baselines, including:

- Multi-domain models: StarGAN [3] and StarGAN* that adds the noise vector into the generator to encourage the diverse outputs.

- Multi-modal models: MUNIT [15], DRIT [22], and *version-c* of SingleGAN [42].

In the above models, MUNIT, DRIT and SingleGAN require a pair of GANs for summer → winter and winter → summer severally. StarGAN-based models and DMIT only use a unified structure to learn the bijection mapping between two domains. To better evaluate the performance of multi-domain and multi-modal mappings, we propose to test inter-domain and intra-domain translations separately.

As the qualitative comparison in Fig. 4 shows, the synthesis of StarGAN has significant artifacts and suffer from mode collapse caused by the assumption of deterministic cross-domain mapping. With the noise disturbance, the quality of generated images by StarGAN* has improved, but the results are still lacking in diversity. All of the multi-modal models produce diverse results. However, without utilizing the style information between different domains, the generated images are monotonous and only differ in simple modes, such as global illumination. We observe that MUNIT is hard to converge and to produce realistic season transfer results due to the limited training data. DRIT and SingleGAN produce realistic results, but the images are not vivid enough. In contrast, our DMIT can use only one unified model to produce realistic images with diverse details for different image domains.

To quantify the performance, we first translate each test image to 10 targets by sampling styles from prior distribution. Then we adopt Fréchet Inception Distance (FID) [12] to evaluate the quality of generated images, and LPIPS (official version 0.1) [44] to measure the diversity [15, 22, 46] of samples generated by same input image within a specific domain. The quantitative results shown in Table 2 further confirm our observations above. It is remarkable that our method achieves the best FID score while greatly surpassing the multi-domain and multi-modal models in LPIPS distance.

Table 2: Quantitative comparison of season transfer.

| | summer→winter | | summer→summer | | winter→summer | | winter→winter | |
|---|---|---|---|---|---|---|---|---|
| | FID | LPIPS | FID | LPIPS | FID | LPIPS | FID | LPIPS |
| StarGAN | 218.78 | - | 233.61 | - | 248.29 | - | 224.37 | - |
| StarGAN* | 152.11 | 0.012 | 135.25 | 0.011 | 153.79 | 0.013 | 149.04 | 0.011 |
| MUNIT | 84.43 | 0.166 | 58.96 | 0.133 | 73.82 | 0.134 | 68.92 | 0.141 |
| DRIT | 58.70 | 0.205 | 49.58 | 0.166 | 53.79 | 0.192 | 57.11 | 0.179 |
| SingleGAN | 63.77 | 0.184 | 51.64 | 0.186 | 54.24 | 0.188 | 57.30 | 0.178 |
| DMIT w/o T-Path | 75.90 | 0.109 | 57.24 | 0.118 | 72.75 | 0.124 | 65.15 | 0.116 |
| DMIT w/o D-Path | 116.71 | **0.545** | 85.97 | **0.513** | 95.63 | **0.517** | 124.96 | **0.544** |
| DMIT w/o $\mathcal{L}_{GAN}^c$ | 60.81 | 0.268 | 43.54 | 0.260 | 50.33 | 0.270 | 58.09 | 0.256 |
| DMIT w/ VanillaD | 63.34 | 0.259 | 44.73 | 0.239 | 50.79 | 0.255 | 60.10 | 0.242 |
| DMIT w/ ProjectionD | 66.50 | 0.289 | 46.92 | 0.301 | 52.4 | 0.293 | 65.66 | 0.299 |
| DMIT | **58.46** | 0.302 | **43.04** | 0.275 | **48.02** | 0.292 | **55.23** | 0.279 |

**Ablation study.** To analyze the importance of different components in our model, we perform an ablation study with five variants of DMIT.

As for the training paths, we observe that both T-Path and D-Path are indispensable. Without T-Path, the model is difficult to perform cross-domain translation as we analyzed in Section 3. In contrast, without D-Path, the generated images are blurry and unrealistic and produce meaningless diversity by the artifacts. Combining these two paths result in a trade-off of quality and diversity of images.

As for the training incentive, we observe $\mathcal{L}_{GAN}^c$ is influential for the diversity score. Without this incentive, the visual styles are similar in summer and winter. It suggests that $D_c$ encourages the model to eliminate the domain bias and to learn well-disentangled representations.

As for the architecture of discriminator, we evaluate two other conditional models with different information injection strategies, including vanilla conditional discriminator (VanillaD) [16, 27] that concatenates input image and conditional information together, and projection discriminator (ProjectionD) [28, 29] that projects the conditional information to the hidden activation of image. The qualitative results in Table 2 indicate that the capacity of VanillaD is limited. The images generated of DMIT with ProjectionD are diverse, but prone to contain artifacts, which leads to its lower FID score. Our full DMIT, equipped style-based discriminator, gets the balance between diversity and quality.

## 4.3 Semantic Image Synthesis

To further verify the potential of DMIT in mixed-modality (text and image) translation, we study on the task of semantic image synthesis. The existing I2I approaches usually assume the state of the domain is discrete, which causes them to not be able to handle this task. We compare our model with the state-of-the-art models of semantic image synthesis: SISGAN [6], Paired-D GAN [26], and TAGAN [30].

Fig. 5 shows our qualitative comparison with the baselines. Although SISGAN can generate diverse images that match the text, it is difficult to generate high-quality images. The structure and background of the images are retained well by Paired-D GAN, but the results do not match the text well. Furthermore, it can be observed that Paired-D GAN cannot produce diversity for conditional input with different samples. TAGAN presents images with acceptable semantic matching results, but the quality is unsatisfactory. By encoding the style from the input image, DMIT can well preserve the original background of the input image and generate high-quality images that match the text descriptions. Meanwhile, DMIT can also produce diverse results by sampling other style representation.

Besides to calculate FID to qualify the performance, we perform a human perceptual study on Amazon Mechanical Turk (AMT) to measure the semantic matching score. We randomly sample 2, 500 images and mismatched texts for generating questions. For each comparison, five different workers are required to select which image looks more realistic and fits the given text. As shown in Table 3, DMIT gets the best of both image quality and semantic matching score. Since retaining the irrelevant information of the input image is important for semantic image synthesis, we also evaluate the reconstruction ability of different methods by transforming the input image with its corresponding text. The scores of PSNR and SSIM further demonstrate the capabilities of our method

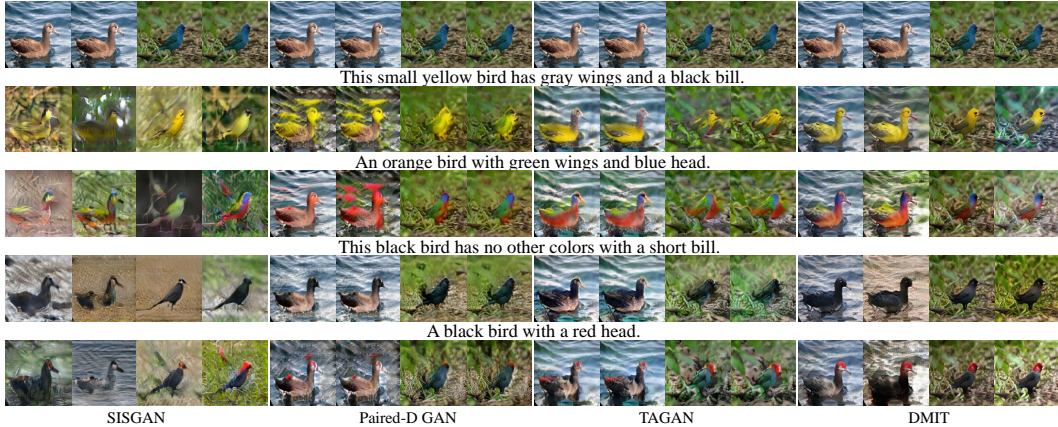

Figure 5: Qualitative comparison of semantic image synthesis. In each column, the first row is the input image and the remaining rows are the outputs according to the above text description. In each pair of images generated by DMIT, the images in the first column are generated by encoding the style from the input image and the second column are generated by random style.

Table 3: Quantitative comparison of semantic image synthesis.

|  | FID | Human evaluation | PSNR | SSIM |
|---|---|---|---|---|
| SISGAN | 67.24 | 15.3% | 11.27 | 0.193 |
| Paired-D GAN | 27.62 | 25.2% | 22.34 | 0.886 |
| TAGAN | 34.49 | 20.4% | 19.01 | 0.736 |
| DMIT | **13.85** | **39.1%** | **25.49** | **0.934** |

in learning efficient representations. It suggests that the disentangled representations enable our model to manipulate the translation in finer detail.

## 4.4 Limitations

Although DMIT can perform multi-mapping translation, we observe that the style representations tend to model some global properties as discussed in [31]. Besides, we observe that the convergence rates of different domains are generally different. Further exploration will allow this work to be a general-purpose solution for a variety of multi-mapping translation tasks.

## 5 Conclusion

In this paper, we present a novel model for multi-mapping image-to-image translation with unpaired data. By learning disentangled representations, it is able to use the advances of both multi-domain and multi-modal translations in a holistic manner. The integration of these two multi-mapping problems encourages our model to learn a more complete distribution of possible outputs, improving the performance of each task. Experiments in various multi-mapping tasks show that our model is superior to the existing methods in terms of quality and diversity.

## Acknowledgments

This work was supported in part by Shenzhen Municipal Science and Technology Program (No. JCYJ20170818141146428), National Engineering Laboratory for Video Technology - Shenzhen Division, and National Natural Science Foundation of China and Guangdong Province Scientific Research on Big Data (No. U1611461). In addition, we would like to thank the anonymous reviewers for their helpful and constructive comments.

## Footnotes

[1]Since encoder $E_d$ has a deterministic mapping, it is no need for joint training with $E_d$ in our training stage.

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
