[Supplementary Material · supplementary.pdf]

# 1 Implementation

## 1.1 Network Architecture

The architecture details are as follows, except that the domain label encoder $E_d$ is organized as a dictionary without learnable parameters. We follow the notations used in [1]: $h$ and $w$: height and width of the input image, $n_s$ and $n_d$: dimensions of style $s$ and domain label $d$, N: the number of output channels, K: kernel size, S: stride size, P: padding size, FC: fully connected layer, IN: instance normalization, LN: layer normalization, CBN: central biasing normalization, LReLu: Leaky ReLu with a negative slope of 0.2.

| Part | Input → Output Shape | Layer Information |
|---|---|---|
| Down-sampling | $(h,w,3) \to (h,w,64)$ | CONV-(N64, K7x7, S1, P3), IN, LReLU |
| | $(h,w,64) \to (\frac{h}{2},\frac{w}{2},128)$ | CONV-(N128, K4x4, S2, P1), IN, LReLU |
| | $(\frac{h}{2},\frac{w}{2},128) \to (\frac{h}{4},\frac{w}{4},256)$ | CONV-(N128, K4x4, S2, P1), IN, LReLU |
| Bottleneck | $(\frac{h}{4},\frac{w}{4},256) \to (\frac{h}{4},\frac{w}{4},256)$ | ResBlock: CONV-(N256, K3x3, S1, P1), IN, LReLU |
| | $(\frac{h}{4},\frac{w}{4},256) \to (\frac{h}{4},\frac{w}{4},256)$ | ResBlock: CONV-(N256, K3x3, S1, P1), IN, LReLU |
| | $(\frac{h}{4},\frac{w}{4},256) \to (\frac{h}{4},\frac{w}{4},256)$ | ResBlock: CONV-(N256, K3x3, S1, P1), IN, LReLU |
| | $(\frac{h}{4},\frac{w}{4},256) \to (\frac{h}{4},\frac{w}{4},256)$ | ResBlock: CONV-(N256, K3x3, S1, P1), IN, LReLU |

Table 1: Architecture of content encoder $E_c$

| Part | Input → Output Shape | Layer Information |
|---|---|---|
| Down-sampling | $(h,w,3) \to (\frac{h}{2},\frac{w}{2},64)$ | CONV-(N64, K4x4, S2, P1) |
| | $(\frac{h}{2},\frac{w}{2},64) \to (\frac{h}{4},\frac{w}{4},128)$ | ResBlock: CONV-(N256, K3x3, S1, P1), IN, LReLu, AvgPool-(K2x2, S2) |
| | $(\frac{h}{4},\frac{w}{4},128) \to (\frac{h}{8},\frac{w}{8},256)$ | ResBlock: CONV-(N256, K3x3, S1, P1), IN, LReLu, AvgPool-(K2x2, S2) |
| | $(\frac{h}{8},\frac{w}{8},256) \to (\frac{h}{16},\frac{w}{16},256)$ | ResBlock: CONV-(N256, K3x3, S1, P1), IN, LReLu, AvgPool-(K2x2, S2) |
| | $(\frac{h}{16},\frac{w}{16},256) \to (256)$ | LReLu, GlobalAvgPool |
| Output Layer($\mu$) | $(256) \to (n_s)$ | FC-(256, $n_s$) |
| Output Layer($logvar$) | $(256) \to (n_s)$ | FC-(256, $n_s$) |

Table 2: Architecture of style encoder $E_s$

| Part | Input → Output Shape | Layer Information |
|---|---|---|
| Bottleneck | $(\frac{h}{4},\frac{w}{4},256)+(n_s+n_d) \to (\frac{h}{4},\frac{w}{4},256)$ | ResBlock: CONV-(N256, K3x3, S1, P1), CBIN, ReLu |
| | $(\frac{h}{4},\frac{w}{4},256)+(n_s+n_d) \to (\frac{h}{4},\frac{w}{4},256)$ | ResBlock: CONV-(N256, K3x3, S1, P1), CBIN, ReLu |
| | $(\frac{h}{4},\frac{w}{4},256)+(n_s+n_d) \to (\frac{h}{4},\frac{w}{4},256)$ | ResBlock: CONV-(N256, K3x3, S1, P1), CBIN, ReLu |
| | $(\frac{h}{4},\frac{w}{4},256)+(n_s+n_d) \to (\frac{h}{4},\frac{w}{4},256)$ | ResBlock: CONV-(N256, K3x3, S1, P1), CBIN, ReLu |
| | $(\frac{h}{4},\frac{w}{4},256)+(n_s+n_d) \to (\frac{h}{4},\frac{w}{4},256)$ | ResBlock: CONV-(N256, K3x3, S1, P1), CBIN, ReLu |
| | $(\frac{h}{4},\frac{w}{4},256)+(n_s+n_d) \to (\frac{h}{4},\frac{w}{4},256)$ | ResBlock: CONV-(N256, K3x3, S1, P1), CBIN, ReLu |
| Up-sampling | $(\frac{h}{4},\frac{w}{4},256) \to (\frac{h}{2},\frac{w}{2},128)$ | DECONV-(N128, K4x4, S2, P1), LN, ReLu |
| | $(\frac{h}{2},\frac{w}{2},128) \to (h,w,64)$ | DECONV-(N64, K4x4, S2, P1), LN, ReLu |
| | $(h,w,64) \to (h,w,3)$ | CONV-(N3, K7x7, S1, P3), Tanh |

Table 3: Architecture of generator $G$. The style $s$ and the domain label $d$ are injected by central biasing normalization.

| Part | Input → Output Shape | Layer Information |
|---|---|---|
| Down-sampling | $(h,w,3) \rightarrow (\frac{h}{2},\frac{w}{2},64)$ | CONV-(N64, K4x4, S2, P1) |
| | $(\frac{h}{2},\frac{w}{2},64)+(n_d) \rightarrow (\frac{h}{4},\frac{w}{4},128)$ | ResBlock: CONV-(N256, K3x3, S1, P1), CBIN, LReLu, AvgPool-(K2x2, S2) |
| | $(\frac{h}{4},\frac{w}{4},128)+(n_d) \rightarrow (\frac{h}{8},\frac{w}{8},256)$ | ResBlock: CONV-(N256, K3x3, S1, P1), CBIN, LReLu, AvgPool-(K2x2, S2) |
| Output Layer | $(\frac{h}{8},\frac{w}{8},256) \rightarrow (\frac{h}{8},\frac{w}{8},1)$ | CONV-(N1, K1x1, S1, P0) |

Table 4: Architecture of discriminator $D_c$ and $D_x$. The domain label $d$ are injected by central biasing normalization.

## 1.2 Training Details

We train all our models with Adam optimizer [3], setting the learning rate of 0.0001 and exponential decay rates $(\beta_1, \beta_2) = (0.5, 0.999)$. To keep each loss close in magnitude, the hyper-parameters are set as follows: $\lambda_{rec} = 10$, $\lambda_{reg} = 1$, $\lambda_{KL} = 0.01$. The batch size is set as one for season transfer and sketch-to-photo tasks, and eight for semantic image synthesis and facial attribute transfer. Besides, we adopt multi-scale strategy proposed by Zhu *et al.* [8] to discriminate the real and fake images in different scales. Since the distributions of content $c$ are still changing, we use the objective of LSGAN [5] to stabilize the training of $D_c$. Besides, we replace the standard adversarial loss of $D_x$ with hinge version [6] to accelerate the convergence. All of the models are trained on a single NVIDIA TITAN V GPU.

## 2 Additional Experiment Results

### 2.1 Season Transfer

Figure 1: **The interpolation on latent representations.** In this experiment, the content representation is extracted from the image of the first row and the style representations are extracted from the first and final columns. The images on the same row have the same style representation and images on the same column have the same domain label representation.

Figure 2: **Example-guided image translation.** In this experiment, the content representation is extracted from the image of the first column and the style representations are extracted from the first row. The images on the same row have the same content representation and images on the same column have the same style representation.

## 2.2 Semantic Image Synthesis

Figure 3: **Semantic image synthesis on CUB [7].** The first row shows the input images. Each of the remaining rows presents the translation results according to the text description.

## 2.3 Facial Attribute Transfer

we perform the facial attribute transfer on the The CelebFaces Attributes(CelebA) dataset [4]. CelebA dataset contains a large number of celebrity images. We preprocess the CelebA dataset according to the method in [1]. Twelve attributes are selected to construct the attribute vector: hair color (black, blond, brown, gray), gender (male/female), age (young/old), expression (with/without smile), and hairstyle (with/without bangs). The quantitative and qualitative comparisons are shown in Table 5 and Fig. 4, and more visual results are presented in Fig. 5.

Table 5: Quantitative comparison of facial attribute transfer.

|  | FID | LPIPS |
|---|---|---|
| StarGAN | 51.20 | - |
| StarGAN* | 48.16 | 0.001 |
| DMIT | **32.36** | **0.066** |

Figure 4: **Qualitative comparison of facial attribute transfer**. The styles of StarGAN* and DMIT are sampled from random noise.

Figure 5: Facial attribute transfer results of DMIT on CelebA (Input, Black hair, Blond hair, Brown hair, Gray hair, Gender, Age, Smile, Bangs). Since "Old" and "Gray hair" are generally present at the same time in CelebA, we thus set the hair to gray to generate a realistic old face.

## 2.4  Sketch-to-Photo

We use the dataset provided by Isola *et al*. [2] to perform Sketch-to-Shoe and Shoe-to-Shoe translations. Our model in this task is trained with unpaired data, and the qualitative results are shown in Fig. 6

Input                  Diverse shoes sampled by our model

Figure 6: **Qualitative results of sketch-to-photo.** The first two rows show the inter-domain translation results of our model, and the last two rows shows the intra-domain translation results.