[Reviews · NeurIPS 2019]

Reviewer 1



Originality - There have been rapid progress in the field of image-to-image translation. Notably, methods generalizes the image-to-image translation to handle unpaired data (e.g. CycleGAN), multi-domain (e.g., StarGAN), and multi-modal outputs (e.g., DRIT, MUNIT). This work builds upon all these methods and presents a method that is capable of performing multi-domain, multi-mapping from unpaired training data. While this is a straightforward and expected extension, I think this is a nice contribution. Quality - The method is technically sound. However, the work integrates existing losses and disentangled representation (e.g., in DRIT, MUNIT) to achieve the multi-domain, multi-output mapping. The technical novelty is somewhat limited. - The experimental validation is not very convincing. 1) The "diversity" has not not quantified. 2) In Table 2, it seems that the main improvement over prior art is due to implementation details (e.g., projectionD). It is hard to draw conclusion from the Table. Also, as far as I know, the LPIPS scores are "the lower the better". Table 2 seems to treat it the other way around. 3) The paper claims that the method can perform "multi-domain" image-to-image translation. Yet, most of the applications shown in the paper (except the face attribute experiments in the supplementary material) are mapping between "two domains" only. Examples include seasons, weather, and time of day. Clarity - The paper is well-written. The implementation details are clearly discussed in the paper and the supplementary material. Significance - I think the paper will stimulate future research in image-to-image translation.

Reviewer 2



-- In Equation 2, it encourages the distribution of styles of all domains to be as close as possible to a prior distribution. However, I am a little confused about how it disentangle different domain styles (E_s_{x_i}). In other words, is there any possibility that E_s_{x_i} and E_s_{x_j}, where i is different from j, are very close? I will appreciate it if the authors can provide more explanations here. -- domain label encoder Ed I suspect the necessity of introducing a domain label encoder here. I think the information extracted from E_{d} can be handled by style encoder E_{s}. I hope the authors can illustrate this in the rebuttal. -- Figure 2 (d) only utilize one generator for multi-modal data, but in previous works for multi-modal translation, which is illustrated in Figure 2 (b), different modal data needs different generators. Why in multi-mapping translation only one generator is enough, as shown in subfigure (d)? -- In Table 2, why DMIT w/o D-Path achieves the best LPIPS score on all the cases (while DMIT outperforms other settings in FID)?

Reviewer 3



The paper unifies the work of the multi-modal and multi-domain translation, each existing separately so far. It provides a novel combination of techniques, so far used to solve each problem alone, in order to solve the unified problem. For example, an adversarial loss is used in the embedding space of the common encoder and a reconstruction loss is used to reconstruct the input from the common, separate and domain information. Such techniques are used to achieved disentanglement for 2 domains only in, for example Munit and [1]. In that sense, the novelty exists in combining those existing techniques from previous works in a clever way. However, each part in isolation (style encoder and embedding, etc) is not novel. The qualitative and quantitative evaluation is very convincing showing that the use of additional domains (and thus additional supervision) can improve disentangled translation. There seem to be a large gap in FID and other scores for semantic image synthesis compared to season transfer. Could the authors comment on why this is? Further, the ablation analysis and comparison to baselines is very thorough. One concern I have is whether the work is able to perform image to image translation between domains where the difference is not only in style. For example, in the celeba domain, considering domains with different attributes (e.g facial hair, glasses, smile). In this case the facial attributes are common to all domains, and the additional attributes (facial hair, glasses smile) is separate. However, since the separate encoder is constructed so as to model only style/global properties, and the additional attributes are content, how would the translation work in this case? Is this a limitation of the work? [1] should be considered for this case. Could the authors comment on any other limitations of the work? With regards to clarify, the paper is very well written and clear. The overview in Fig. 3 is also very clear. As for significance, the work seem important for practitioners, providing superior results to state of the art results so far. For future research, as the underlying ideas used for the solution already exist in previous work (adversarial loss, reconstruction loss, etc), I am not sure if ideas from the work can be built upon and how. Apart from (rather significant) visual and numerical improvements, are there any observations about the results, or new insights that this work provide? [1] Emerging Disentanglement in Auto-Encoder Based Unsupervised Image Content Transfer. Press et al. ICLR 2019.

[Author Response · NeurIPS 2019]

We thank all the reviewers for their helpful comments. We first response to some key concerns and the others will be addressed point by point. **Confusions of the LPIPS metric.** We apologize for the lack of explanation of LPIPS metric. Here LPIPS is the diversity metric that measures the perceptual difference of generated images. This diversity metric was proposed by BicycleGAN and adopted by DRIT and MUNIT. Specifically, we compute the average LPIPS distance between pairs generated by the same input image and different sampled styles. 10 image pairs are generated for each test image. Thus the higher the LPIPS, the better the diversity. We will include the above explanation to improve our work. **The reason for performance improvement.** We would like to clarify that FID and LPIPS are metrics for quality and diversity, respectively. In this work, we argue that learning the styles among different domains results in more diverse sampling space than that learning from one specific domain. From Tab. 2, we observer that DMIT-based models have a significant improvement of diversity over the multi-modal baselines when there is a T-path to encourage cross-domain translation. It suggests that the supervision from multi-domain is beneficial to multi-modal translation. But improving diversity does not result in the improvement of FID, since artifacts may be introduced. Thus the capacity of the discriminator is important for producing realistic images. Please refer to line 216-222 for more analysis. **The performance gap between different tasks.** In season transfer, the main difference between DMIT and baselines is that DMIT aligns the styles among different domains. So there is a significant improvement in diversity. In semantic image synthesis, previous works focus on modeling the foreground and background separately in terms of training losses. Without reasonable representations, these methods are difficult to produce high-quality images. By learning disentanglement, we observe that the style $S$ is associated with background and the content $C$ is related to the foreground. The disentangled representations enable DMIT to perform finer manipulation and achieve better results than the baselines.

**To Reviewer #1: Number of domains.** In addition to facial attribute transfer, semantic image synthesis contains more than two domains as we introduced at line 97-100 and 179-181 of the paper. Since we treat the image set with the same text description as an image domain, there are countless domains. **Compare with StarGAN on CelebA.** As shown in Fig. A (a), all of the methods can produce images that correspond to expected attributes. But the styles of images generated by StarGAN are monotonous, despite the injection of noise vector. The quantitative results also confirm our observation. We will include more comparisons in the supplementary.

**To Reviewer #2: How does Eq.(2) help to disentangle different domain styles?** Eq.(2) encourages to minimize the mutual information of $X$ and $S$ (refer to [1] in the paper). Thus $E_s$ is enforced to model the efficient disentangled representations. Besides, note that we assume the styles among different domains can be aligned (*e.g.*summer nightfall and winter nightfall), which suggests that the representations are domain invariant. To achieve this goal, we utilize a unified (weight sharing) encoder $E_s$ to map images of different domains onto the same space. Thus similar images will have similar representations. But only sharing the mapping function cannot guarantee to eliminate the distribution shift of representations among different domains. Therefore, we encourage the style representations of all domains to be as close as possible to the same distribution to eliminate the domain bias. **Why does DMIT need the encoder $E_d$?** Combined with the above analysis, since we eliminate the domain-specific information of $S$, we need the domain label to indicate the mapping of the target domain. **Why is there only one generator?** Previous methods do not have aligned styles, so they need multiple domain-specific generators. Our method assumes that both $C$ and $S$ can be shared among different image domains, so we can use one generator to perform multi-mapping translation. **Why does DMIT w/o D-Path achieve the best LPIPS score?** Without D-Path, DMIT cannot learn effective representations and produces blurry images. Although the artifacts produce meaningless diversity (LPIPS), the quality of generated images is poor. Without T-Path, DMIT lacks incentives for the use of styles and produces monotonous images that only a subset of real data. Combining both paths allows DMIT to learn effective representations for diverse cross-domain translation.

**To Reviewer #3: Can DMIT perform content transfer?** Yes. We have evaluated DMIT on three additive facial attributes: facial hair, glasses, and smile. As shown in Fig.A (b), we observe that DMIT can add or remove specified facial attributes arbitrarily. **Limitations and future works.** Although DMIT can perform the content transfer, we observe that the style representations tend to model some global properties rather than specific contents, *e.g.*skin color and scene lighting. We agree that the problem is caused by spatial pooling used in $E_s$, as discussed in ContentDisentanglement[1]. To verify the above conjecture, we construct a simple variant of DMIT (DMIT-CD) according to ContentDisentanglement. As shown in Fig.A (c), although there is still room for improvement, DMIT-CD has great potential for multi-domain content transfer. Besides, we observe that the convergence rates of different domains are generally different, *e.g.*adding glasses is more difficult than changing hair color. Thus a domain-adaptive learning strategy may help to improve training stability and performance. We will include these valuable discussions in our work.

Figure A: Visual and quantitative results of facial attribute transfer.

[1]Emerging Disentanglement in Auto-Encoder Based Unsupervised Image Content Transfer. Press et al. ICLR 2019.

[Meta-Review · NeurIPS 2019]

While the reviewer scores diverged a bit, the reviews were actually in good agreement. The reviewers found the paper to be novel and significant because it proposes a novel unified framework multi-domain and multi-modality image to image (I2I) translation (while previous works focus only one of them). However the technical novelty is limited in the sense that this is accomplished by combining losses and disentanglement approaches that have been proposed in the previous works. There were also concerns that the experiments are performed on two domains only, and it is not clear that the method will work between domains where the difference is not only in style. There were also various technical issues that the authors addressed in the rebuttal. After discussion, the reviewers split, with one positive, one marginally positive, and one marginally negative. This made the paper a very borderline case. After discussion between ACs it was decided that the novelty and importance of the problem as well as the soundness of the method justify publication, even though the technical novelty is limited.